plant science

saline-alkali, landscape, photosynthesis, cell structure

**Authors for correspondence:**
Hai-Shun Xu
e-mail: nj_xuhaishun@163.com
Jin-Cheng Xing
e-mail: sdauxxx@163.com

# Growth, physiological and transcriptomic analysis of the perennial ryegrass *Lolium perenne* in response to saline stress

Hai-Shun Xu[1], Su-Ming Guo[1], Lin Zhu[2] and Jin-Cheng Xing[3]

[1]College of Landscape Architecture, Nanjing Forestry University, Nanjing City, Jiangsu Province 210037, People's Republic of China
[2]Design Institution of Wujin Planning and Surveying, Changzhou City, Jiangsu Province 213100, People's Republic of China
[3]Jiangsu Coastal Area Institute of Agricultural Sciences, Yancheng City, Jiangsu Province 224000, People's Republic of China

H-SX, 0000-0002-8780-4815; J-CX, 0000-0002-9947-3304

Salinization of soil is a global environmental concern. To bioremediate or use saline-alkali lands, most studies focused on screening of halophytes and breeding of saline-tolerant non-halophyte vegetables. Seldom studies investigated effects of salinity on general landscape plants, which are important for landscape construction in urban areas. In the present study, effects of salinity on seed germination and seedling growth of the perennial ryegrass *Lolium perenne* were investigated. The final seed germination rate was not affected at salinity up to 6.4‰. Partial seedlings wilted in all saline treatments and the mortality of *L. perenne* was positively correlated with salinity. Treatments with salinity equal to or lower than 1.6‰ did not affect length and dry weight of shoot and root. These results suggested that *L. perenne* could be sowed and then grow well in low-salinity areas. To explore the underlying physiological mechanisms, contents of photosynthetic pigments and antioxidant indices were determined. The results showed that contents of chlorophyll *a*, *b* and carotenoid significantly decreased in all saline treatments, in comparison to the control. Similarly, activities of superoxide dismutase and peroxidase decreased and contents of glutathione and malondialdehyde increased in saline treatments. Additionally, transcriptome analysis identified 792 differentially expressed genes (DEGs) in *L. perenne* shoots between 6.4‰ saline treatment and the control. Compared with the control, genes in relation to iron transportation and amino acid metabolism were

downregulated, but genes participating in energy metabolism were upregulated. These changes would inhibit toxicity of ion accumulation and provide more energy for plants to resist saline stress.

# 1. Introduction

Land salinization is a serious problem and leads to desertification [1]. Saline-alkali soils have important effects on plants. Most plants cannot or hardly survive on saline-alkali soils, since low osmotic potential of soil, imbalanced organic composition and salty toxicity inhibit plant growth and reduce land productivity [2–4]. Due to poor water condition, increased water table, seawater erosion and human activities [5], approximately 954 million hectares worldwide and 99.13 million hectares of lands in People's Republic of China have been salinized [6], which are still increasing every year.

Regarding bioremediation of saline-alkali, most studies focused on utilization of salty soil for agricultural production. To bioremediate saline-alkali lands, hundreds of halophyte species (such as seawater-cultivated vegetables) have been domesticated to reduce soil salinity [7,8] and high-tolerant cultivars of non-halophytes have been bred to adapt to saline-alkali soils [9–11]. These studies paved an effective way to resolve saline-alkali agricultural lands.

With development of economic and living standards, requirements of landscape construction in urban areas are increasing. Worldwide, there are lots of cities built on saline-alkali lands, such as Cangzhou, Yancheng, Daqing and Chifeng in People's Republic of China, the State of Victoria in Australia and California in America. Obviously, green constructions on these cities are quite difficult, since ordinary landscape plants may not be able to survive on the saline-alkali lands. Thus, it is important to screen plant species which are suitable for landscape construction and can tolerate high salinity. Primary studies revealed that *Festuca elata*, *Lolium*, *Poa annua*, *Trifolium repens* and *Phragmites australis* could germinate and/or grow under certain salinity [4,5], suggesting that selection of appropriate plants could promote the landscape construction in saline-alkali cities.

The perennial ryegrass *Lolium perenne* is an important plant, which is broadly used for urban landscape, especially for the construction of pastures. Presently, approximately 250 000 acres of perennial ryegrass are grown in the USA. Perennial ryegrass has strong adaptability to environments. It can grow on a wide variety of soil types. In recent years, planting of perennial ryegrass on saline-alkali lands was also tried. Cui [12] revealed that treatment with 1.5% NaCl did not affect germination and root length of perennial ryegrass. Higher concentrations postponed germination time and decreased germination rate and root length. Qi *et al*. [13] found that treatment with 50 mM NaCl did not affect malondialdehyde (MDA) content in perennial ryegrass, but higher concentrations significantly increased MDA content, suggesting destructive effects of high salinity on cell membranes. Similarly, Liu *et al*. [14] displayed that low concentration of NaCl activated superoxide dismutase (SOD) and peroxidase (POD) activities, but high concentrations decreased them. These results preliminarily explored effects of NaCl on germination, growth and antioxidant responses in perennial ryegrass. However, these results were fragmentary. Systematic investigations concerning effects of salinity on perennial ryegrass are still lacking. Besides, all these studies applied NaCl solution to mimic saline stress. Actually, components of saline-alkali lands are more complicated than pure NaCl solution. Application of NaCl solution could not accurately reveal effects of salinity on plants.

Transcriptome sequencing is a useful technology to explore molecular changes in response to abiotic stress [15]. For example, transcriptomic analysis showed that major transcription factor (TF) (MYB, bHLH, WRKY) and genes related to cell wall loosening and stiffening were regulated in response to saline stress in Bermuda grass (*Cynodon dactylon*) [16]; osmoregulation, serine–threonine protein kinases and ABA-induced genes were upregulated in tall fescue (*Lolium arundinaceum*) under water stress [17]. To the best of our knowledge, there are no reports investigating the transcriptome changes under saline stress in *L. perenne*.

In the present study, to research effects of salinity on the perennial ryegrass, *L. perenne* seeds were germinated and then cultivated in solutions of different salinities, which were prepared by dissolving sea salts in water or Hogland's medium. Germination rate, seedling growth parameters, contents of chlorophyll and antioxidant indices were compared. Furthermore, to investigate the molecular mechanisms underlying saline tolerance in *L. perenne*, shoot and root transcriptomes were profiled and compared between 6.4‰ saline treatment and control. Overall, these results systematically revealed effects of salinity on perennial ryegrass and the underlying molecular mechanisms, which provided theoretical basis for applying *L. perenne* in landscape construction on saline-alkali urban areas.

# 2. Materials and methods

## 2.1. Seed germination

Perennial ryegrass seeds were purchased from Dongshu Seed Company (Shuyang, People's Republic of China). Commercial sea salts were dissolved in deionized water to prepare saline solution at salinities of 0.8, 1.6, 3.2, 6.4, 12.8 and 25.6‰. Deionized water was used as control. For germination assays, seeds were sterilized by soaking in 4% sodium hypochlorite solution for 10 min and then rinsed thoroughly with sterilized water. Three layers of filter paper were placed in 9 cm Petri dishes and then 10 ml of saline solution was added. In each dish, 100 seeds were placed on filter paper. The seeds were germinated in a light humidified incubator at humidity of 80% RH. The light cycle was 12 h : 12 h. At day time, the temperature was 25°C and the light intensity was 4400 lux. At night, the temperature was adjusted to 15°C. The germination experiment was continued for 5 days and germinated seeds were counted every day. Germination rate and germination vigour index were calculated following Manmathan & Lapitan [18]. Each assay was repeated four times.

## 2.2. Seedling culture in saline solutions

Seeds were germinated in deionized water for 6 days. Next, healthy seedlings of similar length (approx. 10 cm) were transplanted to 1/10 Hoagland's solution [19] in plastic pots (20 × 11 × 10 cm). Seedlings were inserted in holes of cystosepiment and then floated on water surface with roots soaked in media. Each pot contained 500 ml of media and 300 seedlings. Seedlings were cultured at 25°C and 70% RH humidity. The light cycle was set as 12 h : 12 h with light intensity of 4400 lux. After 3 days, the culture media were replaced by 1/10 Hoagland's solution with salinities of 0.8, 1.6, 3.2, 6.4 and 12.8‰. 1/10 Hoagland's solution was used as the control. The saline treatments were continued for 18 days. During this period, culture media were changed every 4 days. Finally, all seedlings were harvested. Each treatment included four pots as four independent replicates.

## 2.3. Determination of growth parameters

Ten seedlings from each pot were sampled, rinsed with distilled water and drained using clean blotting paper. Shoot length and root length were measured using a scale. Seedlings with more than 2/3 of shoot length wilting were defined dead. Mortality of seedling was counted. After measurement of fresh weight of shoot and root, samples were completely dried at 70°C for 72 h to determine dry weight.

## 2.4. Measurement of contents of photosynthetic pigments

For each treatment, approximately 200 mg of fresh shoots was completely homogenized with a small amount of $CaCO_3$ and $SiO_2$ powders and extracted in 10 ml of 95% ethanol (v/v) in dark overnight. After centrifugation at $3200g$ for 10 min, debris was discarded and supernatants were adjusted to 25 ml using 95% ethanol. Absorbance at 665, 649 and 470 nm was measured using an UNICO UV-4800 ultraviolet spectrophotometer (Shanghai, People's Republic of China). Contents of chlorophyll $a$ (Chl $a$), chlorophyll $b$ (Chl $b$) and carotenoid (Car) were calculated using the following formulae [20].

$$\text{Chl } a = 13.95 \times A_{665} - 6.8 \times A_{649}$$
$$\text{Chl } b = 24.96 \times A_{649} - 7.32 \times A_{665}$$
$$\text{Car} = \frac{1000\, A_{470} - 2.05 \times \text{Chl } a - 114.8 \times \text{Chl } b}{248}.$$

Four biological replicates were performed for each treatment.

## 2.5. Determination of antioxidant indices

From each pot, 20 seedlings were sampled and then roots and shoots were separated. Approximately 500 mg of root or shoot tissues was homogenized in 4.5 ml of 0.1 M precooled phosphate buffer saline (PBS, pH 7.4) with the assistance of $CaCO_3$ and $SiO_2$ particles. After centrifugation at $4000g$ for 10 min at 4°C, the supernatant was collected and the protein concentration was determined using a Bradford protein assay kit (Tiangen, Beijing, People's Republic of China).

Glutathione (GSH), MDA contents, SOD and POD activities in supernatants were measured using commercial kits produced by Nanjing Jiancheng Bioengineering Institute (Nanjing, People's Republic of China) exactly following the manufacturer's protocols. One unit of SOD activity was defined as the amount of enzyme required to cause 50% inhibition of the reduction of nitro-blue tetrazolium (NBT) in 1 ml reaction solution as monitored at 550 nm. One unit of POD was defined as the amount of enzyme catalysing 1 µg of substrate in the reaction system at 37°C. Four biological replicates were performed for each treatment and each sample was tested in three technical repeats.

## 2.6. Transcriptome sequencing

After treated for 18 days, roots and shoots in the 6.4‰ saline treatment and the control were collected and stored at −80°C for transcriptome sequencing. Total RNA was extracted using Biozol reagent (Hangzhou, China). RNA quality was assayed using Agilent Bioanalyzer 2100 system (Agilent Technologies, CA, USA). Samples with RNA integrity number higher than 8.0 were qualified. RNA concentration was measured using Qubit 2.0 assay. Poly(A)-tailed mRNA was enriched using oligo(dT) magnetic beads (NEB, USA) and then broken into short sequences in the fragmentation buffer. The first strand of cDNA was synthesized using random hexamers and the second strand was obtained using RNase H and DNA polymerase I. The double-stranded DNA was purified using QiaQuick PCR purification kit (Qiagen, Hilden, Germany). After repair of ends using polymerase, DNA fragments were adenylated and ligated to sequencing adaptors (NEBNext, New England Biolabs, MA, USA), the DNA was size-selected by agarose electrophoresis, and fragments of 250–300 bp were amplified by PCR and then sequenced on an Illumina HiSeq X platform to collect paired-end reads. Three biological replicates were performed for each treatment.

## 2.7. Bioinformatics analyses

The quality of raw data was checked and filtered using fastp software [21]. Adaptor sequences, reads having more than five unknown bases and reads shorter than 75 bp were removed. The obtained clean reads were assembled using Trinity (v. 2.3.3.10) program [22], and the longest transcript of each contig was selected as unigene.

The unigene library was annotated by blasting against various databases, including the non-redundant protein (Nr), Interpro and Gene Ontology (GO), Kyoto Encyclopaedia of Genes and Genomes (KEGG), Eukaryotic Orthologous Groups (KOG) and Transcription Factor Database (TFDB). The $e$ values were set as less than $10^{-5}$. The transcriptional level of each unigene was calculated and represented as reads per kilo bases per million reads values (RPKM) using RSEM program v. 1.3.1 [23]. Differentially expressed genes (DEGs) with false discovery rate (FDR) < 0.05 [24] and $|\log_2\text{fold change}| > 1$ were recognized using edgeR [25].

Enrichment analyses of GO and KEGG pathway were conducted using clusterProfiler3 v. 3.8 [26]. The $p$-values were transformed to $Q$-value by setting the FDR $\leq 0.05$ [24]. GO functions or KEGG pathways showing $Q$-value < 0.05 were defined as significant enriched.

## 2.8. Real-time quantitative PCR

In order to validate the transcriptional levels calculated by transcriptome sequencing, eight genes responding to saline treatment were selected for real-time quantitative PCR (RT-qPCR). Glyceraldehyde-3-phosphate dehydrogenase (GAPDH), which showed no significant changes in RPKM values between saline treatment and control, was used as an internal reference gene. Based on the unigene sequences, primers were designed using the Primer-BLAST online tool (https://www.ncbi.nlm.nih.gov/tools/primer-blast/). Primers used in the present study are listed in electronic supplementary material, table S1. qPCR experiments were conducted using BioEasy master mix (Bioer, Hangzhou, China) on a Gene9600 plus qPCR machine (Bioer, Hangzhou, China). To check the specificity of primers, melting curves were conducted after amplification. Results with more than one peak on melting curves were discarded. The transcriptional values were compared by calculating their relative change folds using the $2^{-\Delta\Delta Ct}$ method [27].

## 2.9. Statistical analyses

Normality and homogeneity of variances were detected using the one-sample Kolmogorov–Smirnov procedure and Levene's test, respectively. One-way analysis of variance (ANOVA) was performed to

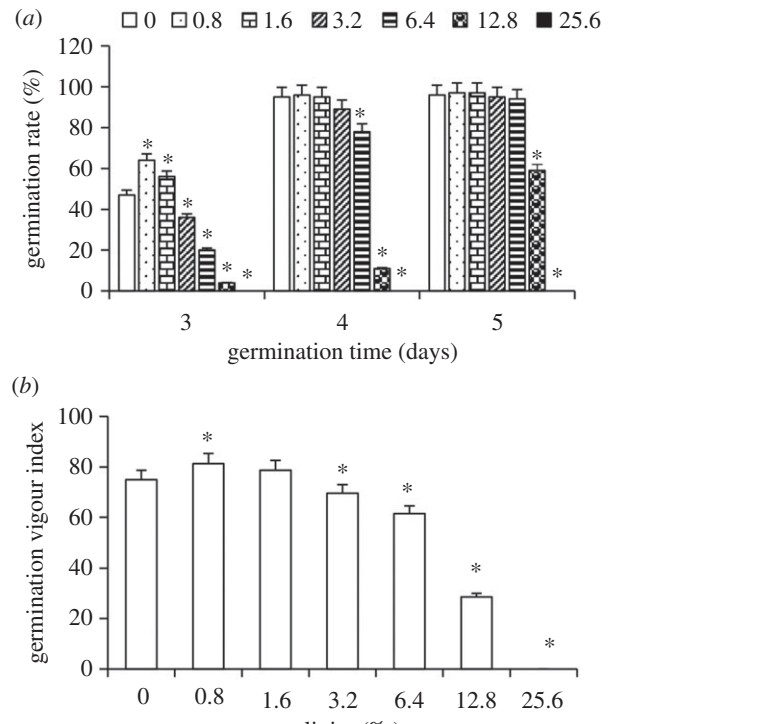

**Figure 1.** (*a,b*) Effects of salinity on seed germination of *L. perenne* (mean ± s.e., *n* = 4). *Significantly different from the control, *p* < 0.05.

check significant effects of salinity on each growth and physiological parameter, followed by multiple comparisons using the least significant difference (LSD) test. Student's *t*-tests were performed to compare the qPCR results between 6.4‰ saline treatment and the control. All statistical analyses were performed using the SPSS 20.

# 3. Results

## 3.1. Effects of salinity on seed germination

Seeds of perennial ryegrass began to germinate from the third day. During the 5 experimental days, no seed germinated at salinity of 25.6‰. At day 3, treatments with 0.8 and 1.6‰ significantly increased germination rate, but treatments with 3.2, 6.4 and 12.8‰ significantly decreased germination rate, compared with the control. Treatments with 6.4 and 12.8‰ for 4 days and treatment with 12.8‰ for 5 days significantly depressed germination rate. No significant differences were observed in other treatments at days 4 and 5, in comparison to the control (figure 1*a*).

Compared with the control, germination vigour index increased in treatment with 0.8‰ but decreased in treatment with salinities ranging from 3.2 to 25.6‰ (figure 1*b*).

## 3.2. Effects of salinity on seedling growth

After cultivation for 18 days, more or less perennial ryegrass seedlings wilted. Mortality increased gradually along with increasing salinity. In all treatments with salinity, mortality was significantly higher than that in the control (figure 2*a*).

The initial seedling length was approximately 10 cm. After cultivated for 18 days, all seedlings grew obviously in all treatments and the control, with shoot length longer than 1.5 cm and root length longer than 8 cm. Compared with the control, shoot length was significantly depressed in treatments with 3.2, 6.4 and 12.8‰; root length was significantly reduced in treatment with 12.8‰ (figure 2*b*). No significant changes were detected in other treatments.

Compared with the control, fresh weight of shoot decreased significantly in treatments with 3.2, 6.4 and 12.8‰, and fresh weight of root decreased significantly at all tested salinities (figure 2*b*). Regarding

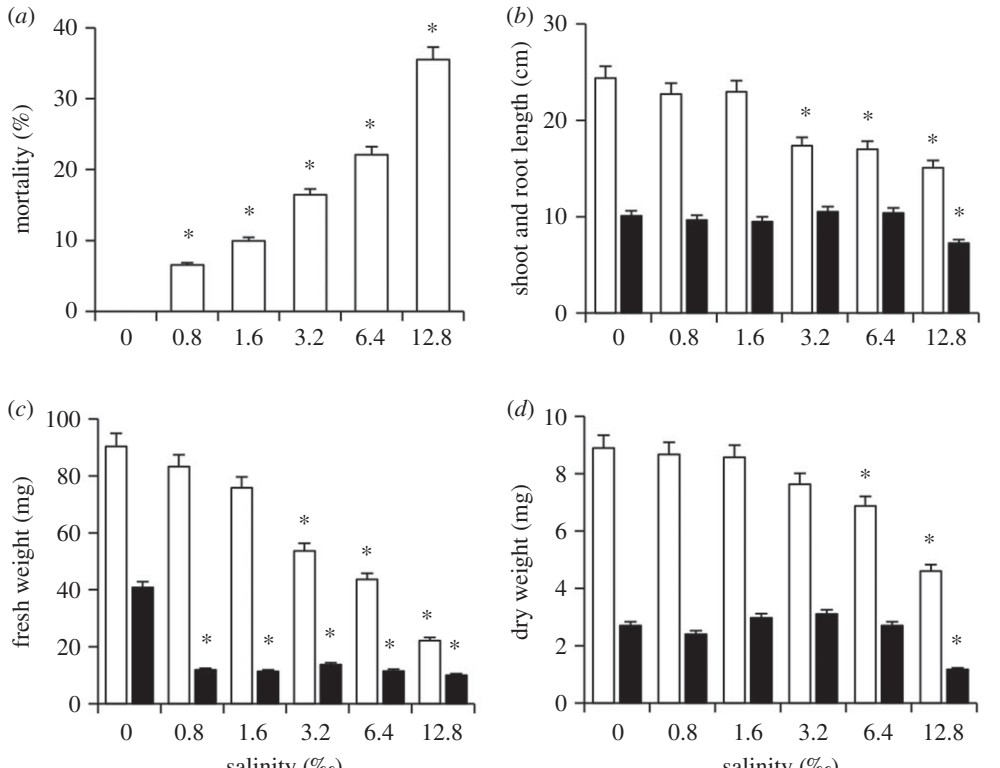

**Figure 2.** (a–d) Effects of salinity on growth parameters of L. perenne (mean ± s.e., $n$ = 10). *Significantly different from the control, $p < 0.05$. White bar, shoot; black bar, root.

dry weight, only treatments with 6.4 and 12.8‰ reduced shoot weight and treatments with 12.8‰ decreased root weight (figure 2d).

## 3.3. Effects of salinity on contents of pigments

Salinity significantly affected contents of Chl $a$, Chl $b$ and Car. In treatments with tested salinities, all these indices were significantly lower than that in the control (figure 3).

## 3.4. Effects of salinity on antioxidant indices

Salinity significantly affected activities of SOD, POD and contents of GSH in both roots and shoots, MDA content in roots but not in shoots. Compared with the control, treatments with 3.2–12.8‰ significantly decreased SOD activity in roots and treatments with 0.8–12.8‰ significantly reduced SOD activity in shoots. POD activity was significantly lower in all salinity treatments in both roots and shoots, compared with the control. Consistently, all salinity treatments increased MDA content in shoots and GSH content in roots (table 1).

## 3.5. Transcriptome sequencing, assembly and functional annotation

The clean data have been deposited in the National Center for Biotechnology Information (NCBI; bioproject numbers: PRJNA596392 for root samples and PRJNA596323 for shoot samples). The total number of unigenes was 158 198 with the average length of 748 bp, GC content of 52.38% and N50 value of 1229 (electronic supplementary material, table S2). Among these unigenes, 48.03 and 3.00% unigenes were shorter than 400 nt and longer than 3000 nt, respectively (electronic supplementary material, figure S1). A total of 45 016 unigenes were predicted as coding sequences (CDSs). Among them, 50.61% CDSs were longer than 400 nt and 16.32% longer than 1000 nt (electronic supplementary material, figure S2). Functional annotation revealed that 158 198 unigenes could be annotated to at least one public database. The 18.41, 26.11, 20.96 and 12.86% unigenes could be annotated to the Nr, Swissprot, KOG and KEGG database, respectively (table 2).

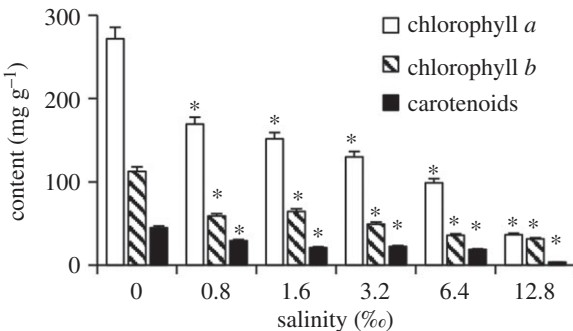

**Figure 3.** Effects of salinity on contents of photosynthetic pigments in *L. perenne* (mean ± s.e., *n* = 3). *Significantly different from the control, *p* < 0.05.

**Table 1.** Effects of salinity on antioxidant indices in root and shoot of *L. perenne* (mean ± s.e., *n* = 4).

| salinity (‰) | SOD (U (mg prot)$^{-1}$) | POD (U (mg prot)$^{-1}$) | MDA (nmol (mg prot)$^{-1}$) | GSH (mg (mg prot)$^{-1}$) |
|---|---|---|---|---|
| roots | | | | |
| 0 | 50.41 ± 1.91 | 45.63 ± 1.62 | 1.42 ± 0.42 | 40.36 ± 6.68 |
| 0.8 | 49.81 ± 4.71 | 29.20 ± 2.23* | 2.16 ± 0.15 | 79.37 ± 8.11* |
| 1.6 | 48.04 ± 2.45 | 20.80 ± 0.69* | 2.54 ± 0.56 | 85.72 ± 10.22* |
| 3.2 | 39.42 ± 2.32* | 26.42 ± 1.49* | 2.79 ± 0.17 | 87.21 ± 5.92* |
| 6.4 | 24.18 ± 3.41* | 19.28 ± 2.07* | 3.51 ± 0.71* | 66.72 ± 8.11* |
| 12.8 | 4.33 ± 2.46* | 21.25 ± 2.04* | 4.99 ± 1.15* | 71.76 ± 6.32* |
| shoots | | | | |
| 0 | 734.5 ± 19.1 | 788.0 ± 64.9 | 5.0 ± 1.9 | 76.3 ± 5.3 |
| 0.8 | 386.7 ± 29.5* | 430.2 ± 43.0* | 6.0 ± 0.4* | 60.2 ± 5.3 |
| 1.6 | 375.9 ± 27.3* | 472.6 ± 33.9* | 6.7 ± 0.3* | 68.3 ± 7.1 |
| 3.2 | 455.5 ± 34.9* | 447.1 ± 35.4* | 9.0 ± 0.4* | 80.5 ± 3.7 |
| 6.4 | 406.0 ± 15.7* | 395.4 ± 19.5* | 8.7 ± 1.0* | 62.5 ± 5.0 |
| 12.8 | 443.4 ± 24.7* | 453.2 ± 14.7* | 11.2 ± 0.6* | 93.0 ± 7.1 |

*Significantly different from the control, *p* < 0.05. mg prot, mg of total protein.

**Table 2.** Number of unigenes annotated based on public databases.

| total unigenes | Nr | Swissprot | KOG | KEGG | overall |
|---|---|---|---|---|---|
| 158 198 | 29 123 | 41 306 | 33 166 | 20 356 | 45 771 |

## 3.6. Differentially expressed genes and qPCR validation

To validate the DEGs identified by transcriptome sequencing, eight DEGs related to saline responses were determined by qPCR. The results showed similar changing tendencies to the relative transcriptional levels calculated by RPKM values (electronic supplementary material, figure S3). Correlation analysis of the ratio of differential expression level from transcriptome to that from qPCR at different salt stress treatments are in good agreement with each other (*r* = 0.777, *p* < 0.05) (electronic supplementary material, figure S4), indicating that the transcriptome data were reliable.

Compared with the control, 10 unigenes were significantly upregulated and four unigenes were significantly downregulated in the saline treatment in roots (figure 4). These unigenes included dehydrin, probable F-box protein At5g04010 and 12 unknown genes (electronic supplementary material, table S3).

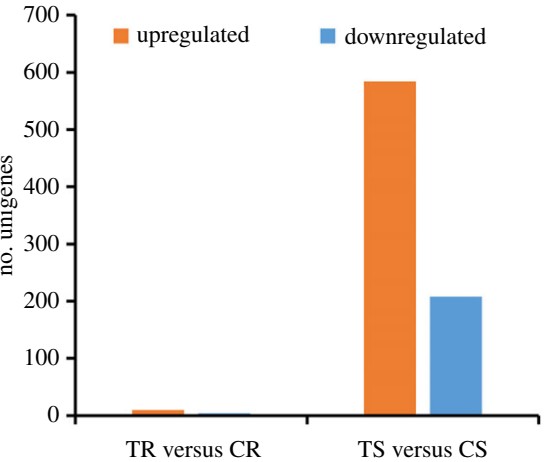

**Figure 4.** DEGs in *L. perenne* roots and shoots in response to saline stress. (T, treatment; C, control; R, roots; S, shoots.)

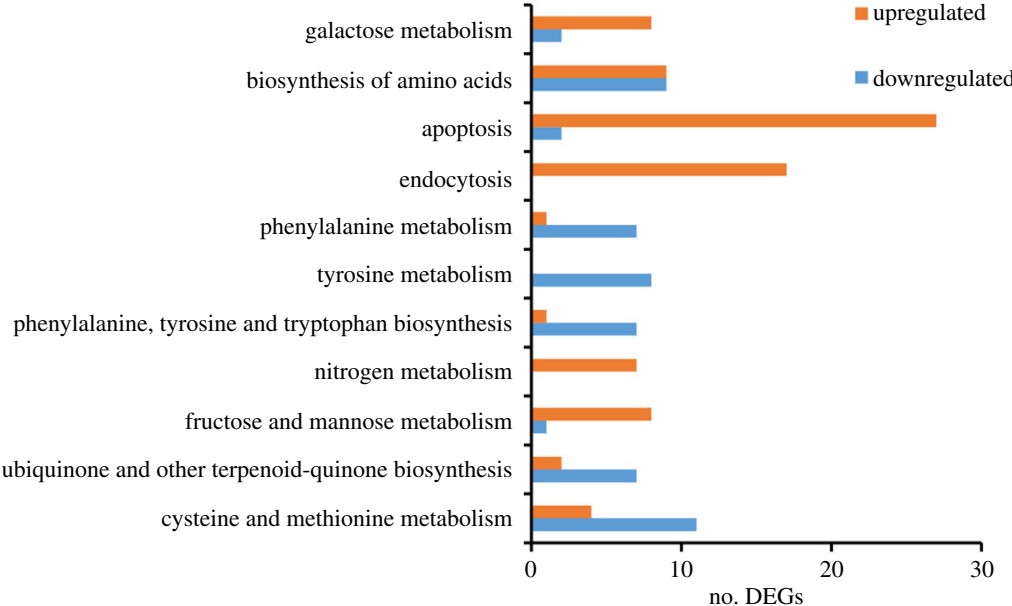

**Figure 5.** KEGG enrichment of DEGs in *L. perenne* shoots.

Compared with the control, 584 unigenes were significantly upregulated and 208 unigenes were significantly downregulated in shoots, respectively (figure 4). GO function analyses showed that upregulated DEGs were categorized mainly in 'catalytic activity', 'binding', 'membrane', 'membrane part' and 'cellular process', while downregulated DEGs were categorized in 'catalytic activity', 'single-organism process', 'cellular process', 'metabolic process' and 'membrane' (electronic supplementary material, figure S5). KEGG pathway analysis significantly enriched 11 pathways ($Q$-value $\leq 0.05$, figure 5). Among these pathways, the upregulated DEGs were enriched in fructose and mannose metabolism, nitrogen metabolism, endocytosis, apoptosis and galactose metabolism. These pathways were mainly involved in energy metabolism. Differently, the downregulated DEGs were enriched in cysteine and methionine metabolism, ubiquinone and other terpenoid-quinone biosynthesis, phenylalanine, tyrosine and tryptophan biosynthesis, tyrosine metabolism and phenylalanine metabolism. These pathways mainly participated in biosynthesis and metabolism of amino acid (figure 5).

## 4. Discussion

Salinity affects seed germination of plants. Most halophytic species germinate to the highest percentages in fresh water, with a rapid decline in germination with increased salinity, but a few species show

remarkable salt-tolerance [28]. Generally, halophytes can tolerate relatively high salinities. For example, germination rate of *Cakile maritima* was not affected in response to treatment with 100 mM NaCl, but was inhibited in treatment with 200 mM NaCl [29]. For non-halophytes, salinity treatments inhibited seed germination. Tolerance of seed germination to salinity varied among species. Treatment with 17 mM NaCl inhibited 68.2% germination rate of *Pteroceltis tatarinowii* [30]. Treatment with 15 mM NaCl did not significantly affect seed germination of *P. tatarinowii*, but higher NaCl concentrations depressed germination rate and seedling growth significantly [30]. *Brassica napus* germination was promoted by treatment with 25 mM NaCl, but treatments with 50 and 100 mM NaCl showed opposite effects [31]. In the present study, germination of *L. perenne* showed similar responses to salinity treatment. Treatment with 0.8‰ promoted germination rate at day 3 and germination vigour index, but treatments with salinity equal to or higher than 3.2‰ depressed seed germination rate and germination vigour index. After 4 or 5 days, germination rate in treatments with 3.2 and 6.4‰ could arrive at the level in control, suggesting that moderate salinity elongated germination time. Low salinity could promote seed germination probably due to the positive effects on deviation and elongation of root cells [32]. Inhibition of high salinity on germination rate and postponed effects of moderate salinity on germination time should have resulted from the elevation of osmotic pressure, which prevented seeds to absorb water [33]. More than half of seeds could finally germinate at salinity up to 12.8‰, suggesting that germination of *L. perenne* could tolerate high salinity.

Under saline conditions, deficiency of water and salt toxicity (sodium and chloride) negatively influence seedling growth and even killed plants [33]. In the present study, wilted seedlings were observed in salinity treatments, but mortality was lower than 20% at salinities up to 3.2‰, suggesting that low salinity did not severely endanger *L. perenne*. Treatments with 0.8 and 1.6‰ did not affect length, fresh weight and dry weight of shoot as well as length and dry weight of root. These results suggested that salinity of 1.6‰ did not affect plant growth and organic matter accumulation in *L. perenne*. In treatments with salinity ranging from 0.8 to 6.4‰, length and dry weight of root were not affected, but its fresh weight decreased significantly, suggesting reduction of water content in root, which should be attributed to high osmotic pressure of culture media. Moreover, length and dry weight of root changed at higher salinity than length and dry weight of shoot did, suggesting that root growth was less sensitive to salinity stress than shoots, which agreed with Munns & Termaat [34].

Contents of photosynthetic pigments were sensitive to salinity stress. In response to salinity treatments, photosynthetic pigments were destroyed [35] and biosynthesis of chlorophylls was inhibited [33]. In the present study, contents of Chl *a*, Chl *b* and Car were all lower in saline treatments than those in the control, consistent with previous reports [36] and also consistent with the observed phenomena of wilted seedlings in the present study. The decreased contents of photosynthetic pigments would block photosynthesis process and then negatively impact seedling height and accumulation of organic matters [37,38].

Under abiotic stress, reactive oxygen species (ROS) were produced in chloroplasts, mitochondria and microbodies during respiration [39], which increased oxidative burden and seriously damaged lipids, proteins and nucleic acids [40]. In response to treatment with salinity, SOD and POD activities increased at low salinity and decreased at high salinity [41]. MDA is produced when polyunsaturated fatty acids in cell membrane undergo peroxidation and could be used as an index of lipid peroxidation [42] and GSH is a low molecular weight tripeptide thiol functioning as an antioxidant. Under salinity stress, salt-sensitive plants generally accumulate more MDA than salt-tolerant plants [43]. In response to salinity treatment, GSH content decreased in salt-sensitive cultivars but increased in salt-tolerant varieties [44,45]. In the present study, MDA content was induced in roots at salinities of 6.4 and 12.8‰ and in shoots at salinities ranging from 0.8 to 12.8‰, displaying lipid and cell membrane peroxidation in *L. perenne*. Activities of SOD and POD decreased in roots and shoots in saline treatments, compared with the control, which might be a sign of plant cell poisoning [46,47]. Reduction of SOD and POD activities also suggested destructive effects of salinity stress on cell machinery in *L. perenne*. These results were consistent with the observed phenomena of wilted seedlings. GSH content was induced in roots but not affected in shoots, which could protect roots from oxidative harms and might explain the less sensitivity of root growth parameters to salinity stress than those of shoots.

Transcriptome is usually used to study molecular responses to abiotic stresses in plants [48]. Dehydrin and F-box protein were significantly differentially expressed in *L. perenne*'s roots following saline treatment. Previous reports showed that the increased expression of dehydrin enhanced the tolerance of plants to low temperature, dehydration, drought, saline and osmotic stress [49,50]. Overexpression of dehydrin in *Arabidopsis* and tobacco increased antioxidant activity and reduced membrane damage. F-box proteins often mediate the mechanism of ubiquitination and degradation of stress-related proteins which may be positive or negative regulatory proteins. The F-box gene directly

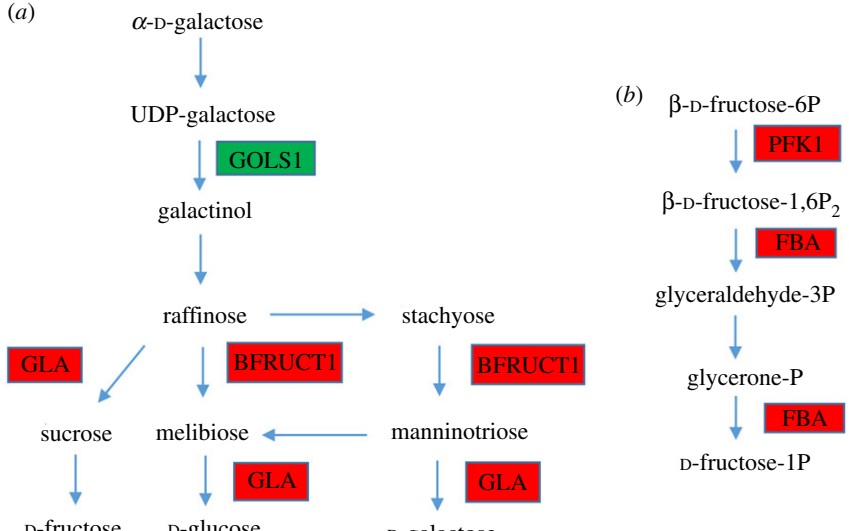

**Figure 6.** DEGs mapped to the energy metabolism pathways in shoots. (a) Galactose metabolism pathway; (b) fructose and mannose metabolism pathway. Red, upregulated with the control; green, downregulated with the control. GOLS1, galactinol synthase 1; GLA, α-galactosidase; BFRUCT1, β-fructofuranosidase 1; PFK1, 6-phosphofructokinase 1; FBA, fructose-bisphosphate aldolase.

or indirectly affects plant hormone signal transduction pathway and affects resistance of plants to saline and alkaline conditions, especially the ABA signal pathway [51,52]. In this study, the expression of dehydrin DHN3 was upregulated and probable F-box protein At5g04010 was downregulated (electronic supplementary material, table S3), indicating that these two genes may be involved in the response of *L. perenne* roots to saline stress.

Saline stress substantially reduced chlorophyll content [53]. A previous study showed that genes encoding enzymes related to chlorophyll synthesis were downregulated under saline stress [38]. The lack of chlorophyll synthetase which catalyses Chlide a or Chlide b to Chl *a* or Chl *b* made the leaves turn yellow [54]. In this study, chlorophyll synthase was significantly downregulated in saline treatment compared with the control (electronic supplementary material, table S4), consistent with Lin *et al.* [38] and the changes of photosynthetic pigment contents in the present study. Activities of antioxidant enzymes decreased in the present study. However, the expression of unigenes encoding antioxidant enzymes, including copper–zinc SOD (Cu/Zn-SOD), manganese SOD (Mn-SOD), iron SOD (Fe-SOD), peroxidase (POD) and glutathione synthetase (GSS) were not significantly different between saline treatment and the control (electronic supplementary material, table S4). These inconsistent results might be attributed to post-transcriptional process.

Ion toxicity is the first response of plants to saline treatment. Generally, $Na^+$ displays more severe influences than $Cl^-$. High concentration of $Na^+$ competes for binding sites with $K^+$ that is essential for cellular functions [55], protein synthesis and activation of enzymes. These changes then would destroy the normal cell process, and water and nutrient absorption [56]. In this study, compared with the control, potassium channel and two-pore potassium channel were significantly downregulated in *L. perenne* shoots under saline stress (electronic supplementary material, table S4), which are major ways of potassium absorption [57]. Similarly, potassium channels including Mkt1, Mkt2 and Kmt1 were found downregulated in the common ice plant (*Mesembryanthemum crystallinum*) under saline stress [58]. These results indicated that the $K^+$ absorption pathways were inhibited in *L. perenne* shoots, which might decrease $Na^+$ absorption and osmotic damages to *L. perenne*.

The energy metabolism of plants is susceptible to saline stress [59]. Changes of sugar content are a common response to environmental stress [60]. Many studies have shown that the sugar content in plants increased under saline stress [61,62]. In this study, three of four DEGs in fructose and mannose metabolism and three of five DEGs in galactose metabolism, which are related to carbohydrate metabolism, were significantly upregulated in response to saline treatment (electronic supplementary material, table S4; figure 6). These upregulated genes included 6-phosphofructokinase 1 (PFK1), fructose-bisphosphate aldolase (FBA) and WRKY transcription factor, which are key genes in the universal glycolysis pathway. PFK1 and FBA catalyse the phosphorylation of D-fructose 6-phosphate and D-Glyceraldehyde 3-phosphate to fructose 1,6-bisphosphate [63], which is the key intermediate of the

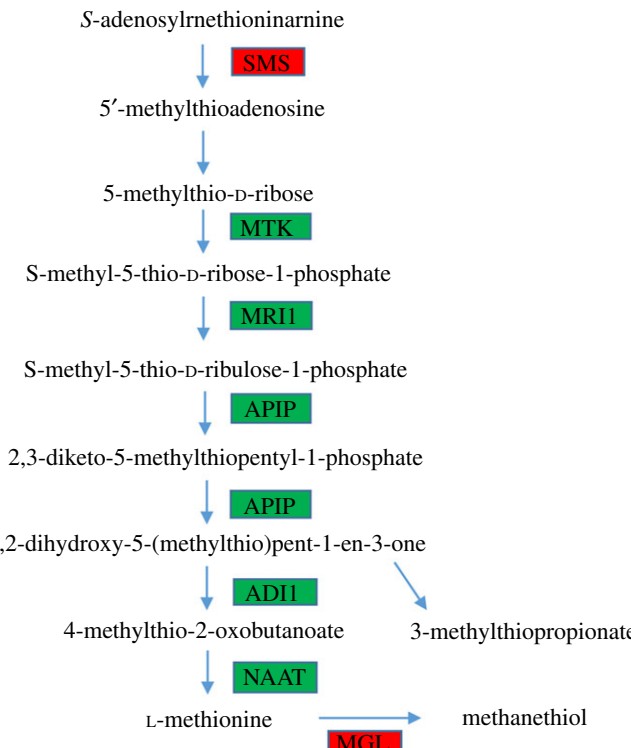

**Figure 7.** DEGs mapped to the cysteine and methionine metabolism pathway in shoots. Red, upregulated with the control; green, downregulated with the control; SMS, spermine synthase; MTK, methylthioribose kinase 1; MRI1, methylthioribose-1-phosphate isomerase; APIP, methylthioribulose-1-phosphate dehydratase; ADI1, 1,2-dihydroxy-3-keto-5-methylthiopentene dioxygenase 2; NAAT, nicotianamine aminotransferase; MGL, methionine gamma-lyase.

glycolysis pathway [64]. Moreover, FBA plays significant roles in saline stress responses [65]. Similar to the present study, saline treatments also upregulated FBA in wheat [66] and rice [67]. Overall, upregulation of these genes indicated higher activation status of glycolysis in *L. perenne* leaves, which would contribute more energy for *L. perenne* to resist saline damages.

Amino acid is the major component of proteins, so amino acid synthesis and metabolism is an important physiological and biochemical process for plant growth and development [68]. In this study, 8 of 13 DEGs involved in amino acid synthesis and metabolism were significantly downregulated in saline treatment, which contributed to cysteine, methionine, tyrosine and phenylalanine metabolism, phenylalanine, tyrosine and tryptophan biosynthesis (electronic supplementary material, table S4). The similar results have been reported in salinity-treated *Suaeda salsa* [63]. Moreover, amino acid metabolism also affects other stress responses in plants [69,70]. Methionine metabolism cycle is the first step of ethylene biosynthesis [71]. In this study, the expression of methylthioribose kinase 1 (MTK), methylthioribose-1-phosphate isomerase (MRI1), methylthioribulose-1-phosphate dehydratase (APIP), 1,2-dihydroxy-3-keto-5-methylthiopentene dioxygenase 2 (ADI1) and nicotianamine aminotransferase (NAAT) were downregulated, while spermine synthase (SMS) and methionine gamma-lyase (MGL) were upregulated (figure 7) in response to saline treatment. Downregulation of methionine metabolism reduced methionine production, which is a source of ethylene [72]. These changes might further influence hormone regulation in *L. perenne* under saline stress. MGL catalyses L-methionine to toxic methanethiol [73]. The upregulation of MGL in this study might accumulate more toxicants in *L. perenne* under saline stress. Further investigations are still required to clarify this issue.

Overall, transcriptome analyses revealed that saline stress downregulated genes in photosynthesis, ion absorption and amino acid metabolism, but upregulated genes in glycolysis in *L. perenne* shoots. These changes might improve resistance of *L. perenne* to salinity.

# 5. Conclusion

The present study revealed that treatments with salinity up to 6.4‰ did not negatively affect seed germination of *L. perenne*. Mortality of *L. perenne* was positively correlated with salinity. Seedling

growth of *L. perenne* could tolerate salinity up to 1.6‰. Higher salinity decreased length and dry weight of root and shoot. The underlying mechanisms included depressed photosynthetic pigments and damages on cell structure. Transcription profile also revealed that the potassium absorption and amino acid metabolism of *L. perenne* were inhibited. The upregulated energy metabolism indicated salt resistance in *L. perenne* under high salinity.

Data accessibility. The clean data have been deposited in the National Center for Biotechnology Information (NCBI) (bioproject numbers: PRJNA596392 for root samples and PRJNA596323 for shoot samples).

Authors' contributions. H.-S.X. carried out the molecular laboratory work, participated in data analysis, carried out sequence alignments, participated in the design of the study and drafted the manuscript; S.-M.G. carried out the statistical analyses and critically revised the manuscript; L.Z. collected field data and critically revised the manuscript; H.-S.X. and J.-C.X. conceived of the study, designed the study, coordinated the study and helped draft the manuscript. All authors gave final approval for publication and agree to be held accountable for the work performed therein.

Competing interests. The authors declare no conflicts of interest.

Funding. This work was supported by Natural Science Foundation for Young Scholars of China (grant no. 51408315), Postdoctoral Science Foundation of China (grant no. 2016M590459), China Scholarship Fund (grant no. 201808320046), the Top-notch Academic Programs Project of Jiangsu Higher Education Institutions (grant no. PPZY2015A063), the Priority Academic Program Development of Jiangsu Higher Education Institution (PAPD) and Youth Science and Technology Innovation Foundation of Nanjing Forestry University (grant no. CX2016003).

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
