## [Reviewer comments · Royal Society Open Science]

Review History

RSOS-200637.R0 (Original submission)

Review form: Reviewer 1

Is the manuscript scientifically sound in its present form?

Yes

Are the interpretations and conclusions justified by the results?

Yes

Is the language acceptable?

Yes

Do you have any ethical concerns with this paper?

No

Have you any concerns about statistical analyses in this paper?

No

Recommendation?

Accept with minor revision (please list in comments)

Comments to the Author(s)

Xu et al. compared the changes of growth and physiological parameters between saline treatments and the control in the perennial ryegrass *Lolium perenne*. The results indicated that *L. perenne* could tolerate certain salinity. Moreover, transcriptomic profiles were investigated and showed that the potassium absorption and amino acid metabolism were inhibited, while the energy metabolism was up-regulated in response to saline stress. As far as I know, there are no reports investigating the transcriptome changes under saline stress in *L. perenne*. This study is intact and contribute significant information. Below are a few suggestions for the authors to make the paper better.

Line 130: "fresh weight" of shoot, root or whole seedling?

Line 134: why did you add CaCO₃ and SiO₂?

Line 135: change rpm to g

Line 139: "the following formula", please add appropriate reference

Line 154: replace manufacture's with manufacturer's

Line 155: define NBT

Line 178: version of Trinity

Line 180: databases

Line 188: Enrichment of GO can be performed using clusterProfiler3?

Line 227: replace elevated with higher

Line 244-245: rewrite as "Salinity significantly affected activities of SOD, POD and contents of GSH in both roots and shoots, MDA content in roots but not in shoots."

Line 253-255: delete. The data availability has been described at the separate section.

Line 275 and 277: "were categorized in" rewrite as "were categorized mainly in"

Line 286: "Figure 4" should be Figure 5

Line 295: please provide the reference

Line 296: replace *Pteroceltis tatarinowii* with *P. tatarinowii*

Line 305: due to

Line 308-310: the font size is inconsistent

Line 321 and 322: replace shoots with shoot

Line 339-340: the font size is inconsistent

Line 367: replace the second "Chlide a" with "Chlide b"

Line 437: PRJNA596323 for shoot samples.

Figure 5 and figure S4: Some texts on the pictures are incomplete.

Review form: Reviewer 2

Is the manuscript scientifically sound in its present form?

Yes

Are the interpretations and conclusions justified by the results?

Yes

Is the language acceptable?

Yes

Do you have any ethical concerns with this paper?

No

Have you any concerns about statistical analyses in this paper?

No

Recommendation?

Accept with minor revision (please list in comments)

Comments to the Author(s)

The current paper of Xu illustrated the effects of salinity on seed germination and seedling growth of the perennial ryegrass *Lolium perenne*. Genes participating in energy metabolism were up-regulated by RNA-seq, which may be contributed to the salt tolerance. The manuscript is well-written and designed. There are some important issues to be resolved as follows.

1. *L. perenne* could be sowed and then grow well in low salinity areas, around 1.6‰ or lower. What about the salt concentration of moderate salinity soil? can *L. perenne* normally grow or germination in high salinity? The salt threshold should be investigated in order to decide whether *L. perenne* was adapted the salinity environment.
2. how many replications were repeat in Transcriptome sequencing?
3. the picture of electrophoresis of SOD and POD should be shown.
4. correlation should be calculated based on qPCR validation and transcriptome.

Decision letter (RSOS-200637.R0)

Dear Dr Xu

On behalf of the Editors, I am pleased to inform you that your Manuscript RSOS-200637 entitled "Growth, physiological and transcriptomic analysis of the perennial ryegrass *Lolium perenne* in response to saline stress" has been accepted for publication in Royal Society Open Science subject to minor revision in accordance with the referee suggestions. Please find the referees' comments at the end of this email.

The reviewers and handling editors have recommended publication, but also suggest some minor revisions to your manuscript. Therefore, I invite you to respond to the comments and revise your manuscript.

- Ethics statement

- Data accessibility

If you wish to submit your supporting data or code to Dryad (<http://datadryad.org/>), or modify your current submission to dryad, please use the following link:
<http://datadryad.org/submit?journalID=RSOS&manu=RSOS-200637>

- **Competing interests**

- **Authors' contributions**

- **Acknowledgements**

- **Funding statement**

Because the schedule for publication is very tight, it is a condition of publication that you submit the revised version of your manuscript before 27-May-2020. Please note that the revision deadline will expire at 00.00am on this date. If you do not think you will be able to meet this date please let me know immediately.

If your manuscript is newly submitted and subsequently accepted for publication, you will be asked to pay the article processing charge, unless you request a waiver and this is approved by Royal Society Publishing. You can find out more about the charges at <https://royalsocietypublishing.org/rsos/charges>. Should you have any queries, please contact openscience@royalsociety.org.

on behalf of Dr Agnieszka Latawiec (Associate Editor)
openscience@royalsociety.org

Associate Editor Comments to Author (Dr Agnieszka Latawiec):

Comments to the Author:

Dear Authors,

The two reviewers have now sent back their comments. Based on their recommendations and my own analysis I suggest the manuscript to be accepted with minor revisions. Please incorporate carefully the suggestions of both reviewers giving particular attention to questions raised by the second reviewer.

Congratulations.

Kind Regards,

Agnieszka Latawiec

Reviewer comments to Author:

Reviewer: 1

Comments to the Author(s)

Xu et al. compared the changes of growth and physiological parameters between saline treatments and the control in the perennial ryegrass *Lolium perenne*. The results indicated that *L. perenne* could tolerate certain salinity. Moreover, transcriptomic profiles were investigated and showed that the potassium absorption and amino acid metabolism were inhibited, while the energy metabolism was up-regulated in response to saline stress. As far as I know, there are no reports investigating the transcriptome changes under saline stress in *L. perenne*. This study is intact and contribute significant information. Below are a few suggestions for the authors to make the paper better.

Line 130: "fresh weight" of shoot, root or whole seedling?

Line 134: why did you add CaCO₃ and SiO₂?

Line 135: change rpm to g

Line 139: "the following formula", please add appropriate reference

Line 154: replace manufacture's with manufacturer's

Line 155: define NBT

Line 178: version of Trinity

Line 180: databases

Line 188: Enrichment of GO can be performed using clusterProfiler?

Line 227: replace elevated with higher

Line 244-245: rewrite as "Salinity significantly affected activities of SOD, POD and contents of GSH in both roots and shoots, MDA content in roots but not in shoots."

Line 253-255: delete. The data availability has been described at the separate section.

Line 275 and 277: "were categorized in" rewrite as "were categorized mainly in"

Line 286: "Figure 4" should be Figure 5

Line 295: please provide the reference

Line 296: replace *Pteroceltis tatarinowii* with *P. tatarinowii*

Line 305: due to

Line 308-310: the font size is inconsistent

Line 321 and 322: replace shoots with shoot

Line 339-340: the font size is inconsistent

Line 367: replace the second "Chlide a" with "Chlide b"

Line 437: PRJNA596323 for shoot samples.

Figure 5 and figure S4: Some texts on the pictures are incomplete.

Reviewer: 2

Comments to the Author(s)

The current paper of Xu illustrated the effects of salinity on seed germination and seedling growth of the perennial ryegrass *Lolium perenne*. Genes participating in energy metabolism were up-regulated by RNA-seq, which may be contributed to the salt tolerance. The manuscript is well-written and designed. There are some important issues to be resolved as follows.

1. *L. perenne* could be sowed and then grow well in low salinity areas, around 1.6‰ or lower. What about the salt concentration of moderate salinity soil? can *L. perenne* normally grow or germination in high salinity? The salt threshold should be investigated in order to decide whether *L. perenne* was adapted the salinity environment.
2. how many replications were repeat in Transcriptome sequencing?
3. the picture of electrophoresis of SOD and POD should be shown.
4. correlation should be calculated based on qPCR validation and transcriptome.

Author's Response to Decision Letter for (RSOS-200637.R0)

See Appendix A.

Decision letter (RSOS-200637.R1)

Dear Dr Xu,

It is a pleasure to accept your manuscript entitled "Growth, physiological and transcriptomic analysis of the perennial ryegrass *Lolium perenne* in response to saline stress" in its current form for publication in Royal Society Open Science.

on behalf of Dr Agnieszka Latawiec (Associate Editor)
openscience@royalsociety.org

Appendix A

Reviewer comments to Author:

Reviewer: 1

Comments to the Author(s)

Xu et al. compared the changes of growth and physiological parameters between saline treatments and the control in the perennial ryegrass *Lolium perenne*. The results indicated that *L. perenne* could tolerate certain salinity. Moreover, transcriptomic profiles were investigated and showed that the potassium absorption and amino acid metabolism were inhibited, while the energy metabolism was up-regulated in response to saline stress. As far as I know, there are no reports investigating the transcriptome changes under saline stress in *L. perenne*. This study is intact and contribute significant information. Below are a few suggestions for the authors to make the paper better.

Line 130: “fresh weight” of shoot, root or whole seedling?

Re: revised.

Line 134: why did you add CaCO₃ and SiO₂?

Re: In order to make the grinding more sufficient, and avoid the destroy of chlorophyll.

Line 135: change rpm to g

Re: revised.

Line 139: “the following formula”, please add appropriate reference

Re: added.

Line 154: replace manufacture's with manufacturer's

Re: revised.

Line 155: define NBT

Re: added.

Line 178: version of Trinity

Re: added.

Line 180: databases

Re: revised.

Line 188: Enrichment of GO can be performed using clusterProfiler3?

Re: ClusterProfiler3 can analyze and visualize functional profiles (GO and KEGG) of gene and gene clusters.

Line 227: replace elevated with higher

Re: revised.

Line 244-245: rewrite as “Salinity significantly affected activities of SOD, POD and contents of GSH in both roots and shoots, MDA content in roots but not in shoots.”

Re: revised.

Line 253-255: delete. The data availability has been described at the separate section.

Re: revised.

Line 275 and 277: “were categorized in” rewrite as “were categorized mainly in”

Re: revised.

Line 286: “Figure 4” should be Figure 5

Re: revised.

Line 295: please provide the reference

Re: added.

Line 296: replace *Pteroceltis tatarinowii* with *P. tatarinowii*

Re: revised.

Line 305: due to

Re: revised.

Line 308-310: the font size is inconsistent

Re: revised.

Line 321 and 322: replace shoots with shoot

Re: revised.

Line 339-340: the font size is inconsistent

Re: revised.

Line 367: replace the second "Chlide a" with "Chlide b"

Re: revised.

Line 437: PRJNA596323 for shoot samples.

Re: revised.

Figure 5 and figure S4: Some texts on the pictures are incomplete.

Re: revised.

Reviewer: 2

Comments to the Author(s)

The current paper of Xu illustrated the effects of salinity on seed germination and seedling growth of the perennial ryegrass *Lolium perenne*. Genes participating in energy metabolism were up-regulated by RNA-seq, which may be contributed to the salt tolerance. The manuscript is well-written and designed. There are some important issues to be resolved as follows.

1. *L. perenne* could be sowed and then grow well in low salinity areas, around 1.6‰ or lower. What about the salt concentration of moderate salinity soil? can *L. perenne* normally grow or germination in high salinity? The salt threshold should be investigated in order to decide whether *L. perenne* was adapted the salinity environment.

Re: In the present study, treatment with 3.2‰ decreased germination rate, shoot and root length. Thus, we considered 1.6‰ might be the highest concentration without obvious effects. Similarly, previous studies revealed that treatment with 4‰ would inhibit the seedling height and dry-matter accumulation, but treatment with 3‰ had no significant effect on seedling growth (Yang and Yang, 2005).

Yang, X. Y. and J. S. Yang (2005). "Effects of Salt Stress on Growth of Ryegrass Seedlings and Its Mitigative Effects of P Fertilizer." *Chinese Journal of Soil Science* 36(6): 899-902.

2. how many replications were repeat in Transcriptome sequencing?

Re: Three replicates were performed.

3. the picture of electrophoresis of SOD and POD should be shown.

Re: We tried to perform electrophoresis of SOD and POD. However, the total proteins have

degraded and can not be used to for electrophoresis.

The present study aimed to investigate physiological changes in response to saline stress. Activity assesses of SOD and POD have been performed and the results indicated that low salinity induced oxidative stress. Thus, our experimental goal has been achieved.

4. correlation should be calculated based on qPCR validation and transcriptome.

Re: added.